# FedGMMAT: Federated generalized linear mixed model association tests

**Wentao Li**[1], **Han Chen**[1,2], **Xiaoqian Jiang**[1], **Arif Harmanci**[1] *

**1** McWilliams School of Biomedical Informatics, University of Texas Health Science Center at Houston, Houston, Texas, United States of America, **2** School of Public Health, University of Texas Health Science Center at Houston, Houston, Texas, United States of America

\* arif.o.harmanci@uth.tmc.edu

**Data Availability Statement:** All data and code are available on GitHub repository at link https://github.com/Li-Wentao/FedGMMAT. Alzheimer's Disease Genotype-Phenotype data can be accessed from Database of Genotypes and Phenotypes (dbGAP)

## Abstract

Increasing genetic and phenotypic data size is critical for understanding the genetic determinants of diseases. Evidently, establishing practical means for collaboration and data sharing among institutions is a fundamental methodological barrier for performing high-powered studies. As the sample sizes become more heterogeneous, complex statistical approaches, such as generalized linear mixed effects models, must be used to correct for the confounders that may bias results. On another front, due to the privacy concerns around Protected Health Information (PHI), genetic information is restrictively protected by sharing according to regulations such as Health Insurance Portability and Accountability Act (HIPAA). This limits data sharing among institutions and hampers efforts around executing high-powered collaborative studies. Federated approaches are promising to alleviate the issues around privacy and performance, since sensitive data never leaves the local sites. Motivated by these, we developed FedGMMAT, a federated genetic association testing tool that utilizes a federated statistical testing approach for efficient association tests that can correct for confounding fixed and additive polygenic random effects among different collaborating sites. Genetic data is never shared among collaborating sites, and the intermediate statistics are protected by encryption. Using simulated and real datasets, we demonstrate FedGMMAT can achieve the virtually same results as pooled analysis under a privacy-preserving framework with practical resource requirements.

## Author summary

Traditional GWAS approaches require the transfer of sensitive genetic data to a central location for analysis, raising privacy concerns and data security issues. We propose a federated learning technique named FedGMMAT to conduct large-scale genetic analyses without the need to share sensitive information. By allowing data repositories to perform local computations and only share the aggregated results in encryption, FedGMMAT ensures that individual genetic data remains secure and private. This novel algorithm enhances the capability of researchers to collaborate on GWAS across multiple sites, thereby increasing the statistical power and robustness of genetic discoveries while maintaining the confidentiality of participant data. This approach not only addresses privacy

with accession identifier phg000049.v2. This dataset is protected under dbGAP data usage agreements imposed by the data owners. Accession must be obtained via dbGAP (https://www.ncbi.nlm.nih.gov/gap/). To access this dataset, users must obtain an eRA-Commons account (https://public.era.nih.gov/commons/) and login to dbGAP with these credentials. In dbGAP, users can request accession to phg000049.v2 by starting a new project, filling out the application form including a data usage agreement, project description, and a signing official. The accession may require an Institutional Review Board approval (IRB) letter. The dataset phg000049.v2 can be selected at the requested data selection step.

**Funding:** XJ is CPRIT Scholar in Cancer Research (RR180012), and he was supported in part by Christopher Sarofim Family Professorship, UT Stars award, UTHealth startup, the National Institute of Health (NIH) under award number R01AG066749, R01LM013712, and U01TR002062, and the National Science Foundation (NSF) #2124789. AH was supported by startup funds from The University of Texas Health Science Center Houston and NIH Grant R01HG012604. The funders had no role in study design, data collection and analysis, decision to publish, or preparation of the manuscript.

**Competing interests:** The authors have declared that no competing interests exist.

concerns but also paves the way for larger scale genetic studies by enabling the participation of diverse data sets from various institutions.

# 1 Introduction

With the surge of genomic data generated by Next Generation Sequencing (NGS) [1–4], number of available genomes have surpassed millions. These data provides opportunities to map genetic factors underlying complex diseases and phenotype. Arguably, Genome-Wide Association Studies (GWAS) are the most popular methods for uncovering the genetic determinants of phenotypic variation [5–7]. State of the art GWAS tools rely employ mixed modeling to assess the statistical significance between variant genotypes and phenotypes while accounting for the biases in ancestry and cryptic relatedness [8–11]. Among these methods, linear mixed models (LMMs) have been popularly used for analyzing continuous traits, their intrinsic assumptions (uniform distribution of residuals with respect to variant allele frequency) may bias analyses of the binary traits. For these cases, generalized linear mixed models have been shown to provide unbiased results [12].

Although statistical modeling has advanced in recent years, the real utility of data can be only realized when large sample sizes and more diverse populations are available to identify salient genetic variants. However, it is challenging to build centralized repositories of large datasets mainly due to privacy concerns and data usage agreements [13]. Numerous studies demonstrated that genetic data is very identifying of its owner and their families [14–20]. Due to very high dimensional nature of genomic data (i.e., very large number of variants), sharing even summary statistics, and other related intermediate data, or simply the existence of variants (e.g., beacons [21–23]) can create risks of membership inference [24], i.e., whether an individual with known genotypes participated a study. These studies have led to an increased public proclivity against sharing genomic data, related intermediate statistics, and an increased protection of GWAS datasets. Although these restrictions have been partially relieved by NIH in 2018 [25–28], open sharing of raw genotype-phenotype data is still not allowed. In parallel, sharing of data from underserved, historically isolated, and vulnerable populations (Ashkenazi Jewish community, Havasupai Tribe) is challenging since these populations are more vulnerable for stigmatization and further isolation [29–31]. Due to these concerns, the genetic and biomedical data analysis and research is even more challenging among these communities [32]. Further concerns are related to unauthorized usage of genetic data for solving cold cases [33], usage of genomic data for probing and potential risk of diverting genealogy databases [34].

With the increasing concerns about misusing genome data, regulations like Health Insurance Portability and Accountability Act (HIPAA) and General Data Protection Regulation (GDPR) have been proposed and implemented to protect individual health records including genetic data [35]. The privacy concerns make it untenable to perform, high-powered and high-quality GWAS because these studies require very large sample sizes [36–38]. As a result, genome data repositories such as UK Biobank [3] often release data with strong restrictions on how data can be used with no possibilities of sharing. Besides the limited data accessibility, many more healthcare data are siloed and underutilized due to concerns around privacy. Some initiatives must be taken to connect these distributed data islands and fully use them while protecting privacy.

Federated Learning (FL) is a method to bridge isolated data silos without sharing the actual data. Such a property can help comply with the local privacy regulations to provide means to

perform collaborative GWAS across data silos. Although some related research has been proposed to provide secure GWAS, most of them focus on adopting federated methods in $\chi^2$ statistics test [37, 39], Principal Component Analysis (PCA) [40–42], and linear/logistic regression tests [39, 41]. Among these approaches, there is currently a lack of focus on correcting the confounding by random polygenic effects while performing association testing, mainly due to computational complexity.

The polygenic effects stem mainly from clustering or correlations between the input subjects. The most well-known case scenario is the existence of known and unknown family members in the case or control cohorts. Since family members share DNA segments among each other, the participation of related individuals into GWAS may shift the allele frequencies and create a confounding bias resulting in false positive signals. In addition, these members are expected to have correlated phenotypes when the studied phenotypes are highly polygenic. It has been shown in previous studies that the [43, 44] that unknown or cryptic relatedness may cause noticeable biases and must be corrected in GWAS. The most popular approach for correcting the confounding by kinship is to introduce an additive polygenic term that is distributed as a zero-mean multivariate Gaussian whose covariance matrix is proportional to the genetic relatedness matrix (or kinship matrix) among the subjects (S1 Fig).

In this paper, we present a privacy-preserving federated GWAS algorithm using generalized linear mixed effects model, FedGMMAT, that adopts an efficient and flexible two-step score testing [12, 45, 46]. FedGMMAT integrates homomorphic encryption (HE) and a symmetric One-Time Pad (OTP)-like encryption mechanism to protect intermediate statistics efficiently while being aggregated in the federated setting. Sensitive genotype and covariate datasets are not shared among the computing entities. We use real and simulated datasets to demonstrate the accuracy and practicality of FedGMMAT in terms of resource requirements.

The source code and example data is available in GitHub repository https://github.com/Li-Wentao/FedGMMAT.

## 2 Results

### 2.1 FedGMMAT algorithm

In the basic setup of FedGMMAT consists of the collaborating sites with local phenotype, genotype, and covariate datasets. Each site also have access to the design matrices that are used to define the random effects among the individuals. For modeling the polygenic effects, the genetic kinship matrix is used to parametrize the covariance of the random effect. The kinship matrix can be computed using a secure kinship estimation method [32]. Each site gains access to the partition of the kinship matrix that quantify kinship to its subjects to all other subjects. A central server is used for aggregating intermediate statistics in the federated protocols. Central server is assumed to be a trusted computing entity (no collusions) and is responsible for setting up asymmetric (homomorphic encryption keys) [47] and symmetric keys for encrypting matrices using a one-time pad (OTP)-like encryption (S2 and S3 Figs, Methods) and performing the aggregation operations in the protocols. Central server has access to the full design matrices (i.e., full kinship matrix from all sites). To protect intermediate steps, FedGMMAT combines symmetric encryption with partitioned noise matrices (Methods, S2 and S3 Figs) to flexibly and efficiently protect intermediate statistics in the computation heavy portions of the algorithm.

We assume that Central Server is trusted and it adheres to the details of the protocols. The main risks entail hacking of Central Server and accidental leakage of intermediate matrices, which should be protected in protocols.

After the keys are shared among all sites, FedGMMAT first executes the null model fitting using a round-robin schedule among the sites wherein each site locally updates the model parameters, encrypts the intermediate results using homomorphic encryption and passes them to the next site to be securely aggregated. The final results are passed to the central server which decrypts the aggregated matrices, updates the global model parameters, and shares them with all sites. It should be noted that null model fitting does not utilize any of the sensitive genotype data. After model parameter convergence, FedGMMAT fits the mixed effect model parameters using a similar round-robin algorithm. The final stage of FedGMMAT is assignment of the score-test statistics to each variant. In this step, sites work with the central server to calculate the score statistics and its variance for all genetic variants. In this step, the aggregation operations make use of OTP-like encryption rather than the HE-encryption to protect matrices to decrease computational and network usage. The final p-values are assigned using the score test statistics at each site.

FedGMMAT makes extensive usage of encrypt-aggregate-decrypt-partition mechanism to securely aggregate and send partitions of intermediate statistics among sites. Since this mechanism only requires additions, lightweight symmetric OTP-like protections help with efficiently protecting the intermediate matrices. FedGMMAT also uses a site-specific partitioned noise matrices (Methods, S2 and S3 Figs) that are necessary to protect the plaintext intermediate matrices after they are decrypted at Central Server using respective decryption keys. Thus, the sites first add the partitioned noise matrices followed by the encryption via OTP-like encryption.

## 2.2 Experimental setup

We designed 3 different experiments to evaluate our proposed FedGMMAT method. We also perform a larger scale experiment to evaluate the resource requirements of different mechanisms in FedGMMAT.

1. The first experiment is on a synthetic dataset in the baseline model from the R package 'GMMAT'. This dataset contains 400 samples and 100 SNPs with features of age, sex, and outcome. To mimic the federated learning settings, we randomly split the 400 samples into 3 distinct datasets with sample sizes 124, 120, and 156.

2. The second experiment is on two synthetic genotype datasets that consider population Homogeneity and Heterogeneity. Homogeneous genotype dataset was simulated using the 1000 Genomes Project data using Central Europeans living in Utah (CEU) population by sampling of variants randomly for each subject. For simulating kinship among the subjects, we used a 4-level 16-member pedigree containing 8 founder members and 8 descendants [32]. In the simulation, the founders' genotypes are sampled, which were used to probabilistically simulate descendant genotypes. We first generated 400 pedigrees (6,400 subjects), which were subsampled down to 6,000 subjects. The overall $6,000 \times 6,000$ kinship matrix was estimated using SIGFRIED [32]. For simulating the phenotypes, we selected 20 random causal SNPs and assigned random effect sizes to each from normal distribution $N(0, 0.5)$. Environmental effect size was randomly sampled from $N(0, 0.5)$ for each individual and gender effect size was set to fixed level of 0.1. Phenotype information was simulated evaluating a logistic link function on the weighted linear combination of covariates and genetic effects on each individual. Population covariates for both homogeneous and heterogeneous datasets were estimated by projection of the genotype data onto 1000 Genomes reference panel [48]. For heterogeneous sample, the genotype data was generated from 3 populations (GBR, YRI, MXL). Both synthetic datasets comprise 6,000 subjects with 62,375 SNPs and 6

covariates (5 population-level PCs and gender). We also preprocessed the genotype data by removing SNPs with minor allele frequency (MAF) less than 0.01, the homogeneous dataset has 56,478 SNPs left and the heterogenous dataset has 62,250 SNPs left. To mimic the federated learning settings, we randomly split the 6,000 samples into 3 sites. Each site has a sample size of 1973, 2037, and 1990 respectively.

3. The third experiment is on 2,545 samples with 571,135 SNPs and 5 covariates (4 PCs and gender). The data is derived from dbGaP (accession identifier phg000049.v2 under restricted access) with Alzheimer's disease as the phenotype. After filtering out SNPs with MAF less than 0.01, we have 551,062 SNPs. To mimic the federated learning settings, the 2,545 samples are randomly split into 3 distinct datasets with samples size 825, 835, and 885.

## 2.3 Concordance of variant association

We first compared the concordance between FedGMMAT and GMMAT under the scenarios described previously.

**2.3.1 Experiment 1 (400 samples in R package 'GMMAT', 100 SNPs).**   First, we evaluated the performance of FedGMMAT by using the same synthetic datasets in R package 'GMMAT'. The synthetic dataset includes 400 samples and 100 SNPs. The baseline model in R will fit a GLMM with age and sex as covariates and disease status as the outcome.

The FedGMMAT first created a privacy-preserving federated network for 3 isolated trainers and trained a federated GLMM model with the 3 split datasets with isolated computation environments. The coefficients of fixed-effects and mixed-effects of GMMAT and FedGMMAT are shown in Table 1.

We observed a very high concordance between the estimates of fixed-effects and mixed-effect coefficients when GMMAT and FedGMMAT are compared. Fig 1 shows the differences of P-values in SNPs' score test between GMMAT and FedGMMAT with log-scale. The plot shows all SNPs' p-values of GMMAT and FedGMMAT are virtually identical (Table 2), laying in the diagonal line.

**2.3.2 Experiment 2 (6k synthetic samples, 62k SNPs).**   Next, we compared the results of FedGMMAT with the synthetic homogeneous and heterogeneous datasets. For the homogenous dataset, the samples are from one population (EUR) while the heterogeneous dataset has multiple populations (AFR, EUR, ASIAN). We include the comparison of the model performance based on the two different settings to test the robustness of our proposed method (Table 2). Similar to the federated settings in Experiment 1, we randomly split the dataset into 3 isolated trainers. Since the homogeneous and heterogeneous settings are sharing the same covariates data, the null models are the same under the two different settings. The model coefficients between GMMAT and FedGMMAT are in Table 3. The scatter plot of the p-values in the score test with synthetic is shown in Fig 2.

**Table 1. Coefficients of GMMAT and FedGMMAT in experiment 1.**

|  | fixed-effect | | | variance component |
|---|---|---|---|---|
|  | **Intercept** | **age** | **sex** |  |
| GMMAT | 0.4721 | -0.0068 | -0.0864 | 0.3377 |
| FedGMMAT | 0.4721 | -0.0068 | -0.0864 | 0.3377 |

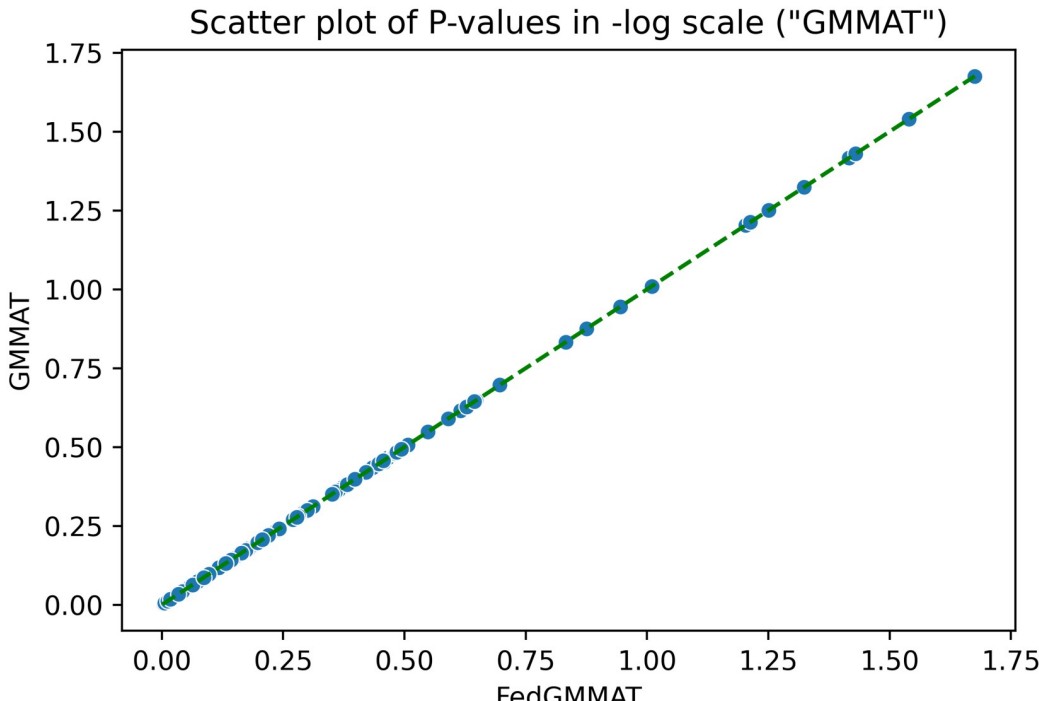

**Fig 1. Scatter plot on -log10 scale of P-values of Synthetic data in FedGMMAT and GMMAT (Experimental Setup 3).**

**2.3.3 Experiment 3 (2.5k dbGaP samples, 551k SNPs).** We finally compared the results from GMMAT and FedGMMAT (Table 2) under the real data usage (dbGaP LOAD Dataset). Among the 3,007 samples with Alzheimer's disease status, we removed the subjects with unknown phenotype labels, which yields the a dataset with 2,545 subjects who were genotyped at 551,062 SNPs after quality control. In this experiment, we also show the capability of our proposed method with 3 federated trainers. The model coefficients can be seen in Table 4, and the scatter plot of P-values is shown in Fig 3.

**Table 2. P-values estimation accuracy of FedGMMAT.**

|  | Covariates | Samples | SNPs | Spearman's correlation | Pearson correlation |
|---|---|---|---|---|---|
| "GMMAT" | 2 | 400 | 100 | 1.0 | 1.0 |
| Heterogeneous | 6 | 6,000 | 62,250 | 0.9999 | 0.9996 |
| Homogeneous | 6 | 6,000 | 56,478 | 0.9683 | 0.9477 |
| Real data | 5 | 2,545 | 551,062 | 1.0 | 1.0 |

**Table 3. Coefficients of GMMAT and FedGMMAT in experiment 2.**

|  | fixed-effect | | | | | | | variance component |
|---|---|---|---|---|---|---|---|---|
|  | Intercept | gender | PC1 | PC2 | PC3 | PC4 | PC5 |  |
| GMMAT | -0.0850 | 0.1215 | -0.0011 | -0.0056 | 0.0120 | 0.0238 | -0.0096 | 0.5959 |
| FedGMMAT | -0.0850 | 0.1215 | -0.0011 | -0.0056 | 0.0120 | 0.0238 | -0.0096 | 0.5958 |

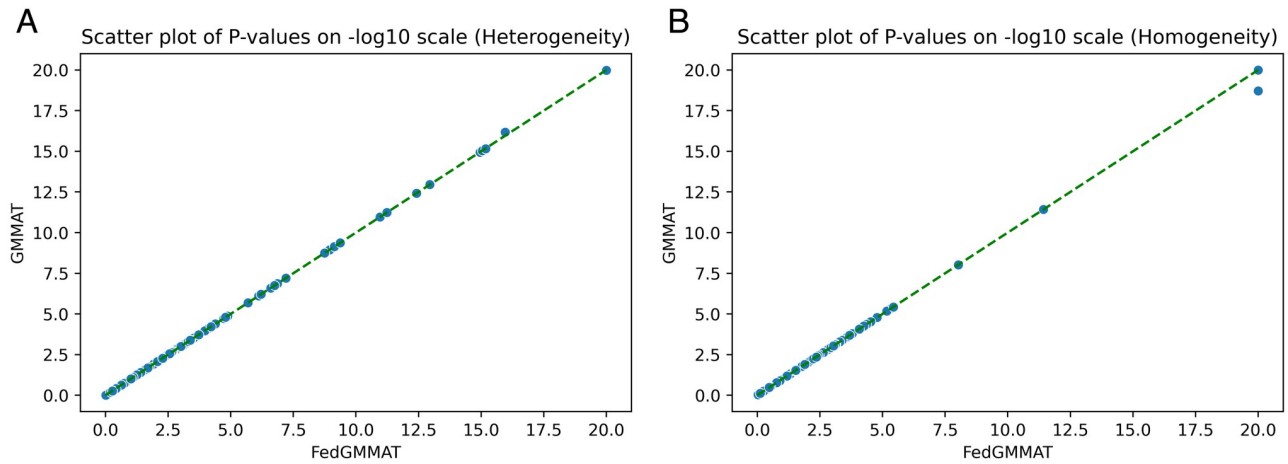

**Fig 2. Scatter plot on -log10 scale of P-values in FedGMMAT and GMMAT.** (A) Heterogeneity; (B) Homogeneity.

**Table 4. Coefficients of GMMAT and FedGMMAT in experiment 3.**

| | fixed-effect | | | | | | variance component |
|---|---|---|---|---|---|---|---|
| | **Intercept** | **gender** | **PC1** | **PC2** | **PC3** | **PC4** | |
| GMMAT | -0.0038 | 0.0854 | -0.1076 | -0.1316 | -0.0321 | 0.2205 | 0.3997 |
| FedGMMAT | -0.0038 | 0.0854 | -0.1076 | -0.1316 | -0.0321 | 0.2205 | 0.3997 |

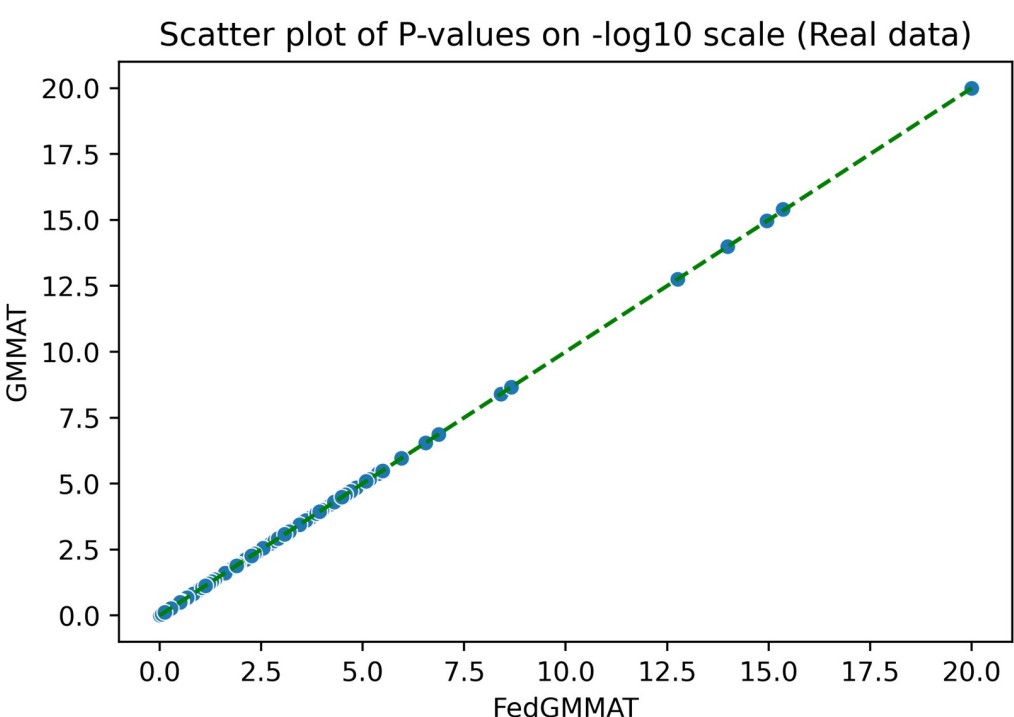

**Fig 3. Scatter plot on -log10 scale of P-values in FedGMMAT and GMMAT on real data.**

## 2.4 Communication cost performance

Last, we demonstrate the communication cost of our method. We compared the computation time between non-HE protection FedGMMAT and HE protection FedGMMAT. Under the experiment of 6k samples, 62k number of SNPs, and 3-site federated learning simulation, we found the size of the SNPs subset that sequentially feeds in the score test can largely affect the total computation time. The comparison of computation time between HE and non-HE in FedGMMAT can be shown in Fig 4.

To perform a more realistic simulation of the computational cost of running FedGMMAT, we simulated 32,000 subjects with 1 million SNPs as a separate test case scenario. For simulating the subjects, we used 2,000 16-member pedigrees of 3 generations with 8 founders and 8 descendants [49]. The simulations were performed using three European populations (GBR, TSI, FIN). We used 11 covariates (1 random assigned gender and 10 population PCs) and the binary phenotype is simulated using 5000 randomly selected variants with effect sizes sampled from $N(0, \sigma = 0.5)$. Random environmental noise standard deviation of 0.5. We divided the dataset among 3 sites and executed FedGMMAT on the same computer (96 core Intel Xeon Platinum 8168 CPU clocked at 2.7GHz) such that each site is run in order to simulate round-robin steps. This is a fair test since the computation heavy protocol comprises round-robin-based updates to parameters. We assume that the sites use an AWS-based cloud system where shared volumes are mounted on instances that are running FedGMMAT among sites for efficient sharing of intermediate datasets. This can be accomplished using Amazon Web Services' EBS multi-attach, which is a very efficient way of sharing data in the cloud environment.

We tested two versions of FedGMMAT that uses homomorphic encryption (HE) in each step and one without HE (referred to as non-HE), which represents the computational

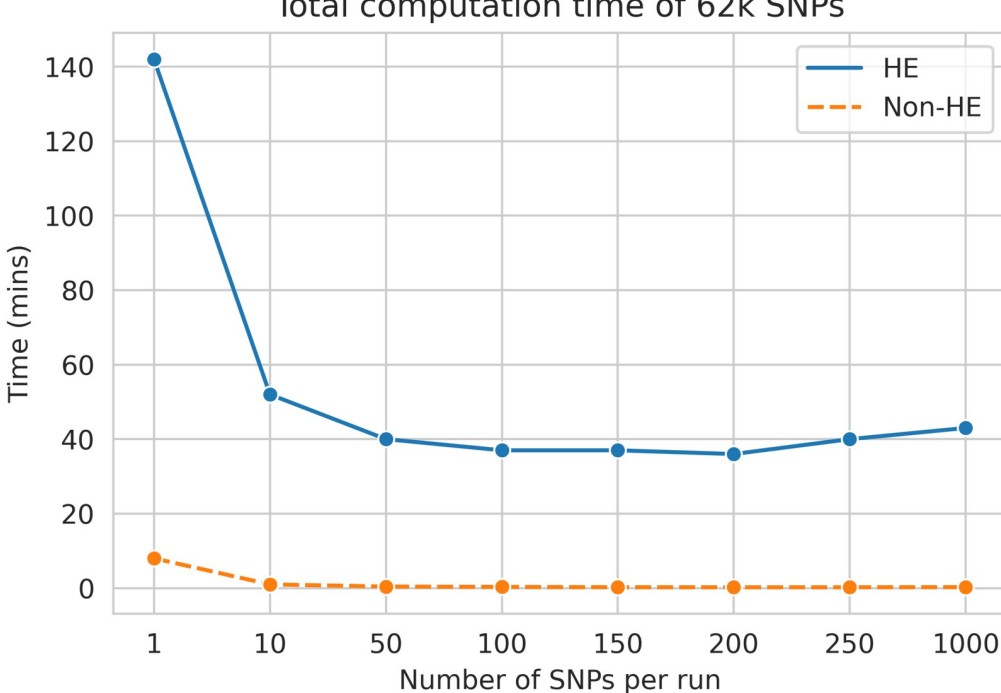

**Fig 4. The comparison of computation time with different numbers of SNPs per batch in minutes between HE and Non-HE protection of FedGMMAT.**

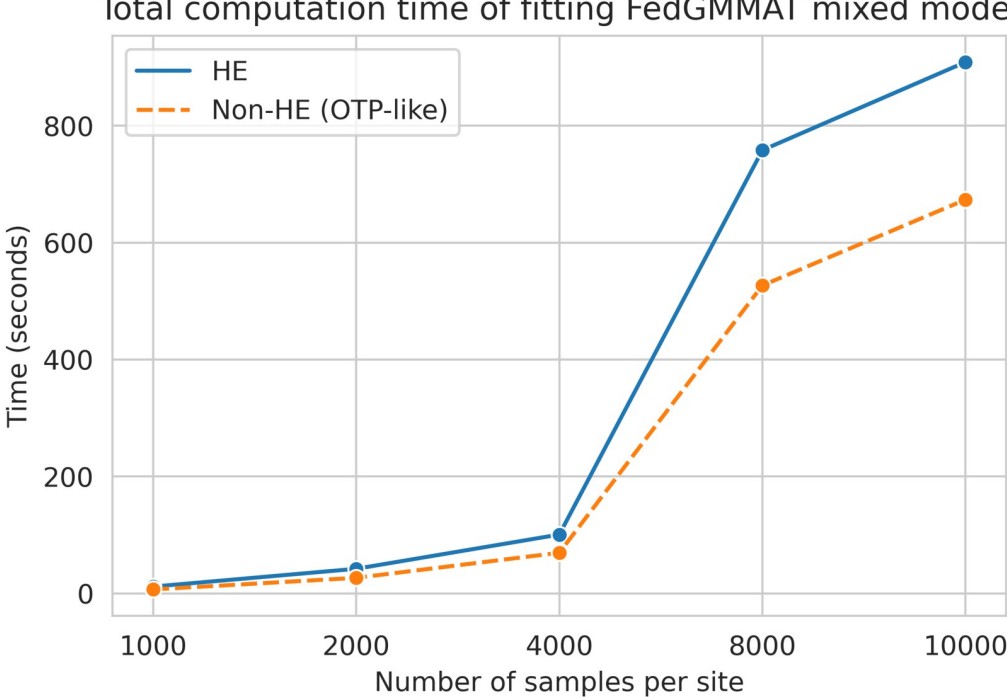

**Fig 5. The total run-time per site (y-axis) in second with respect to the number of subjects on each site (x-axis).**

requirements of the OTP-like encryption scheme that is used for protecting data among sites for score testing stage of FedGMMAT algorithm. Overall, OTP-like (non-HE) algorithm completed in 2 days, 16 hours, 18 minutes and HE version completed in 5 days, 10 hours, 31 minutes. These results indicate that there is moderate-to-high computational requirements for FedGMMAT at large scale datasets.

To evaluate which steps take the most amount of time in HE-based calculations, we measured the scaling of run-time in null model fitting step. We observed even with HE, the model fitting takes around 15 minutes (Fig 5), which indicates that virtually the whole runtime is spent in calculating the score test for the 1 million variants. Given that there is a large difference between HE and OTP versions, we reasoned that efficient security measures of OTP-like approach benefits the efficiency, while the same trust model as the HE-based protocol among the sites wherein the central server is assumed to be trusted by the sites to not deviate from the protocol.

Furthermore, the network requirements of this approach does not entail ciphertext expansion (approximately 12–15 time larger in comparison to the original matrix sizes) that is observed in HE-version (to be described below). We therefore expect that the OTP-like method can substantially increase performance for large scale calculations with FedGMMAT by circumventing the usage of HE at the scoring step. Furthermore, the current testing scenario adheres strictly to using round-robin-based aggregations, which can be parallelized by performing aggregation in on site, which sends the data to Central Server to be decrypted and processed. We leave such optimizations to future studies as they relate to optimizing the underlying protocols without major changes.

<u>Network and Storage Requirements:</u> We next focused on quantifying the total network requirements of FedGMMAT on the large simulated dataset. We divided the total data transfer

**Table 5. Network traffic from each client to the server.**

| | Client to server | Data size | Non HE (OTP-like) encryption memory size (MB) per client | HE memory size (MB) per client |
|---|---|---|---|---|
| Null model | $(\sum_{j=1}^{J} \tilde{\ell}_j^{(t)})$ | $p$-dim vector | 4.577E-5 | 0.0344 |
| | $(\sum_{j=1}^{J} \mathbf{H}_j^{(t)})$ | $p \times p$ matrix | 0.0005 | 0.4124 |
| Mixed model | $(\sum_{j=1}^{J} \mathbf{\Xi}_j)$ | $p \times p$ matrix | 0.0010 | 0.4125 |
| | $(\sum_{j=1}^{J} \mathbf{\Omega}_j)$ | $n \times p$ matrix | 2.9297 | 38.7689 |
| | $(\hat{\boldsymbol{b}}_j)$ | $n$-dim vector | 0.2441 | 3.2999 |
| Score test | $(\mathbf{G}_j^{\top}(\mathbf{P}\tilde{\mathbf{Y}})_{(j)})$ | $k$-dim vector | 7.6294 | 103.1208 |
| | $G_j^{\top} P_j^{\top}$ | $k \times n$ matrix | 48.828125 (k = 200) | 644.6279 (k = 200) |
| | $GP_jG_j$ | $k \times k$ matrix | 0.3052 (k = 200) | 4.1245 (k = 200) |

size for each step of the model fitting and score testing steps for HE and non-HE versions (Tables 5 and 6). non-HE version is included as a proxy for the OTP-like encryption version.

Network Usage for Null Model Fitting: In total, the server-client uses 61 gigabytes of data transfer for fitting the null model with 12 iterations of the model fitting stages. The most bandwidth is spent exchanging $\Sigma^{-1}$ and $V\Sigma^{-1}$ matrix partitions among sites.

Network Usage for Scoring Test: The HE and non-HE (i.e., OTP-like) versions require approximately 3.66 terabytes and 359.2 gigabytes, consecutively. The highest network transfer is used by $\mathbf{G}^{\top}\mathbf{P}$ matrix passed from clients to the server. Of note, the OTP-like encryption strategy allows us to decrease the network requirements further.

Quantization of Intermediate Matrices: One approach to improve performance (memory usage, network usage) is to quantize large matrices. For example, in the scoring stage, we can quantize the $\mathbf{G}^{\top}\mathbf{P} + \mathbf{N}^{(\mathbf{part})} + \mathbf{N}^{(\mathbf{server})}$ matrices from 64-bit precision down to 32-bit precision before they are sent for aggregation when OTP-like protection scheme is used. This approach effectively halves the storage and network requirements. To test quantization of $\mathbf{G}^{\top}\mathbf{P}$ matrix in scoring, we quantized $\mathbf{G}^{\top}\mathbf{P}$ matrices to 32-bits and ran scoring of randomly selected 200-variant chunk. The final p-values of the variants matched with more than 0.999 Pearson R2 correlation to the original results. This result indicates that the scoring stages can be run with quantized matrices using OTP-like protection of the matrices. At the null model fitting, similar quantization can be applied to the scoring stage where partitions of $\mathbf{\Sigma}^{-1}$ and $\mathbf{V\Sigma}^{-1}$ are sent over to the clients, which take up the most amount of space. We tested quantizing these matrices to 32-bit representations (i.e. float32 type) and observed that there is again virtually no loss of accuracy in the predicted null model parameters compared to 64-bit representations.

**Table 6. Network traffic from server directly to the clients.**

| | Server to clients | Data size | Non HE (OTP-like) memory size (MB) |
|---|---|---|---|
| Null model | $\boldsymbol{\alpha}^{(t)}$ | $p$-dim vector | 9.1553E-5 |
| Mixed model | $(\mathbf{V\Sigma}^{-1})_{(j)}$ | $n \times n_j$ matrix | 2604.1667 |
| | $\Sigma^{-1}$ | $n \times n_j$ matrix | 2604.1667 |
| | $\boldsymbol{b}^{(t)}$ | $n$-dim vector | 0.2441 |
| | $\hat{\tau}$ | scalar | 3.8147E-6 |
| Score test | $(\mathbf{P}\tilde{\mathbf{Y}})_{(j)}$ | $n_j$-dim vector | 0.0817 |
| | $P_j$ | $n \times n_j$ matrix | 2604.1667±11.9809 |
| | $GP_j$ | $k \times n_j$ matrix | 16.2760±4.1245 (k = 200) |

Usage of Cloud Resources can improve performance with little cost overhead. We recommend using cloud-based storage among collaborating institutions where sites can cross-mount shared volumes (Using, for example, Amazon Web Services' EBS multi-attach) for sharing of the large intermediate datasets. If cloud usage is not viable, our results indicate that the HE-based null model fitting followed by the scoring strategy that uses OTP-like protection with $\mathbf{G}^\top\mathbf{P}$ matrices quantized to 32-bits can be accomplished in approximately 210 Gbytes of data transfers per client (30.5 Gbytes for model fitting and 179.6 Gbytes for scoring test), which we believe can be practically run over high speed networks among institutions using a shared disk space [50] between institutions.

## 2.5 Comparison to other methods

Numerous methods have been developed that focused on building generalized mixed effects model for biomedical data analysis. We review and compare some of these methods here. Yan et al. [51] developed a method for analyzing EHR datasets with a focus on building small scale models. This method represents an important contribution although We could not immediately identify a way to adopt this method for performing a GWAS using the additive mixed effect models phenotype used in statistical genetics.

Similarly, Li et al. [52] developed a method to detect pleiotropy, i.e., a single variant impacting multiple phenotypes using results from Phenomewide Association Studies (PheWAS) studies. This study is more related to a meta-analysis approach, which makes use of the summary statistics from multiple sites. On the other hand FedGMMAT represents a mega-analysis approach that processes the raw datasets.

To address this problem, we recently developed dMEGA [53]. dMEGA enables performing federated GWAS on a smaller scale (several thousand variants) using a single random intercept parameter. In comparison, FedGMMAT represents a more realistic setting for modeling phenotype using the additive mixed effects. In particular, dMEGA cannot readily model polygenic effects and has more computational requirements compared to FedGMMAT.

Recently Chen and Edupalli et al. [54] developed a linear mixed model for performing federated GWAS analysis. This method provides a federated approach for application of linear mixed models. These approaches are effective for modeling continuous phenotypes. However, for the binary case/control studies, these linear mixed models are not effective since they may fail to capture the heteroskedasticity of the binary phenotype (i.e., variance-mean relationship) [12]. Indeed, GMMAT method was specifically designed and developed to mitigate these limitations of the linear mixed models and we are focusing on this problem for this specific purpose in developing FedGMMAT. Secondly, to our understanding, the described in Chen-Edupalli et al.'s [54] method can only handle random effects that stem from familial relationships, i.e., polygenic effects quantified by dense genetic relatedness matrix. It does not immediately provide the means of integrating other relationship information (e.g., shared environmental factors, longitudinal study designs).

On the other hand, Chen-Edupalli et al's [54] approach does not require explicit calculation of the genetic relatedness matrix (GRM) and is more advantageous compared to FedGMMAT from this perspective. However, when the GRM matrix is not explicitly calculated, there may be limitations when admixed populations exist in the cohorts [55] because the implicit assumptions of GRM integration into the method may not appropriately dissect the effects of confounding by population and cryptic kinship in admixed individuals. Overall, both FedGMMAT and Chen-Edupalli et al's [54] method provide their own set of advantages and we believe they are important contributions to the literature that different use cases can benefit from.

Finally, Zhu et al. [56] developed a federated generalized linear mixed model to tentatively perform federated GWAS analysis [56]. This method by Zhu et al. [56] is one of the first methods to perform federated generalized mixed effects testing. We have extensively tested this algorithm in our previous study (dMEGA) and found that it does not report p-values for the genetic variants; it only reports the effect size estimates. Furthermore it does not model the polygenic random effects stemming from cryptic relatedness effects. Finally we also observed that Zhu et al.'s [56] method may exhibit poor convergence due to utilizing an MCMC-based approach.

## 2.6 Privacy and security of federated protocol

We discuss the privacy and security of FedGMMAT's federated protocol by analyzing information exchanges to present the limitations that can warrant future research. Overall, FedGMMAT use homomorphic encryption and combine it with an OTP-like symmetric protection mechanism that uses site-specific noise addition to protect individual-level and intermediate information between sites and also from the central server. Sites must ensure that the symmetric keys (the secret random seeds from the server) are protected against leakage. Similarly, decryption key must be kept securely at the Central Server.

Central Server is assumed to be a trusted entity that executes the protocols as agreed upon (e.g., [57], [50]), without the necessity of storing identifying information from the sites (i.e., does not need to store registration and identity information). Since Central Server is a trusted authority, the main concern regarding Central Server is the event of hacking or accidental leakage of the datasets from the server. We discuss these cases below.

The central server gains access to, $\tilde{y}$, a proxy for the binary phenotypes (without other identifiers), the kinship matrix, and the null model parameters. Revealing kinship matrix to the Central Server may or may not create privacy concerns for the sites as described in their data usage agreements. In essence, a kinship matrix is a proxy for measuring relatedness between individuals. While it provides some indication of familial pedigrees, it cannot immediately be used to build familial lineages without other information (i.e., an individual's half-sibling and aunt have same kinship coefficient with the individual, similarly individual's niece (or nephew) and uncle have same kinship coefficient with the individual. Thus it is not reliable for uncovering family pedigrees to match to existing pedigrees. The kinship matrix depends on the subjects in the cohort and is not a fixed re-identifying information, i,e., the rows (or columns) of the kinship matrix is not a unique signature for each individual that is invariant between different kinship matrices that harbor the same individual. Also, most kinship matrices are extremely sparse with the exception of the familial studies, which include a uniform selection of families (e.g., trios). Thus, we assume that the kinship matrices are not of immediate concern even if they are leaked unless other re-identifying information are leaked with them. One exception are datasets with families comprising large numbers of children, which may by themselves be re-identifiable due to being outliers. These extreme outlier sized families can be: 1) kept out of the studies or 2) can be considered with families of similar sizes only or 3) they can be subsampled and used in a proxy GWAS setting.

As an alternative to using the full kinship matrix, the sites can further evaluate the amount of cross-sites relatedness and choose to not include cross-site kinship into calculations. This way, the full kinship matrix does not need to be shared among sites. This can be a reasonable assumption in many situations when collaborating sites are geographically distant, which implicitly renders the extent of cryptic kinship across sites very small. In this case, the sites can calculate $\mathbf{\Sigma^{-1}}$ and $\mathbf{V\Sigma^{-1}}$ without minimal exchanges by exploiting the block diagonality of the kinship matrix.

$\tilde{y}$ is sent to the Central Server while variance component ($\tau$) is being updated and at the beginning of the score statistic assignment at the second stage of FedGMMAT. The main concern around leaking of $\tilde{y}$ is combining it with kinship matrix. Although a curious entity may be able to build partial family trees and assign phenotypes, it is not immediately clear how these can be used for re-identifying individuals since most families will be of similar size with no immediate way to distinguish them in terms of phenotypic patterns. Furthermore, the entities do not know the phenotype that is being studied (this information does not have to be revealed). Finally, $\tilde{y}$ is a vector that is used in two places in FedGMMAT and it can be re-submitted and discarded after they are used.

The null model parameters are also revealed to the central server. These include Hessian ($p \times p$) and gradients ($p$ long vector) used to obtain the initial estimates of $\alpha$; updates to the covariate fixed effect vector $\hat{\alpha}$ ($p$ long vector) and the scalar variance component scale $\tau$. Overall, these matrices are calculated using the dense covariate information (no the genotype information), which is low dimensional features from subjects. For a GWAS with large covariate set of 20 features, the total size of covariate-related data that revealed to the central server is on the order of 500. Even though these values are updated over multiple iterations, they are highly correlated between different iterations. Also, the Newton's method is known to down-weight the Considering that there are are thousands of subjects in the GWAS study, we deem the risk fairly low to re-identify an individual. However, we do acknowledge extreme outlier cases may be rarely re-identifiable and a systematic evaluation of re-identification in realistic scenarios is needed to estimate this risk. We recommend removing extreme outlier cases with respect to covariate information. Notably, the cases which are covariate outliers are already excluded by default in GWAS analysis pipelines to ensure they do not bias the analysis results [58], but this should be performed with care to make sure power is not affected by removal of subjects.

Of note, current mechanisms can be used to protect $\tilde{\mathbf{y}}$ in the protocols that requires it to be shared: To update tau, the central server can calculate ($\mathbf{Y}'\mathbf{Z}\mathbf{Y} - \mathbf{P}\mathbf{V}$) where $\mathbf{Z} = \mathbf{P}\mathbf{V}\mathbf{P}$ (Protocol for Fitting Mixed Effects Null Model in S4 Fig). Notice that calculation of $\mathbf{Y}'\mathbf{Z}\mathbf{Y}$ is very similar to $\mathbf{G}'\mathbf{P}\mathbf{G}$ (Protocol for Calculating Score Variance in S5 Fig) and using a similar protocol. This mechanism requires exchanging the horizontal partitions of $(\mathbf{P}\mathbf{V}\mathbf{P})_{(\mathbf{j})}$ matrix among sites. Secondly, $\tilde{y}$ is used in calculating score statistics for all SNPs (Protocol for Calculating Score Statistic in S5 Fig) as $\mathbf{T} = \mathbf{G}'\mathbf{P}\mathbf{Y}$. This quantity can be calculated by first pooling $\mathbf{G}'_{\mathbf{j}}\mathbf{P}_{(\mathbf{j})}$, and then pooling $(\mathbf{G}'\mathbf{P})_{(\mathbf{j})}$ where $(\mathbf{G}'\mathbf{P})_{(j)}$ is the vertical partitioning of $\mathbf{G}'\mathbf{P}$ matrix.

## 3 Discussion

In this paper, we proposed a federated genetic association testing algorithm, FedGMMAT. FedGMMAT is flexible to integrate multiple subject-level random effects and can be used in different study designs without explicit changes to the algorithmic setup and formulations. This is an inherent advantage that stems from the flexible formulation of the efficient score testing of GMMAT. We also believe that the two-stage significance estimation is advantageous from a privacy preserving perspective because the overall problem is modularized in specific steps of null model fitting and variant scoring where covariates and genotype information is used in different stages. In the future, this can in turn enable a tiered privacy-enabling design for GWAS such that different information is protected at different levels as required by local regulations.

It should be noted that there can be inherent challenges in federated studies. For instance, these studies must be designed and planned way ahead of time to collect harmonized and uniform data at independent sites with matching covariates and phenotypes. Due to these challenges, the sites may choose to perform meta-analyses by sharing only the summary statistics,

which requires much less computational burden on the sites with low privacy concerns. However, previous studies have shown that meta-analysis may lose substantial power when there is heterogeneity among the sites. It is therefore important for sites to have the incentives to perform the federated analysis.

Future work can improve several aspects of FedGMMAT. Currently, the null model information is sent in unprotected form among the sites. However, it should be noted that FedGMMAT's null model contains very small number of parameters (on the order to 10–20) and we foresee that it is unlikely that the global model parameters can lead to reliable membership or reconstruction attacks.

FedGMMAT assumes an honest-but-curious adversarial model, which is the predominant assumption for the adversaries in genomic data analysis. The central site must be a trusted and non-colluding entity. This can be achieved by setting up a central key management and aggregation service that is operated by a trusted entity (e.g. NIH). Of note, the central server performs lightweight operations and does not bear substantial computational load, which can be implemented as a trusted service. Future studies are necessary to ensure that the malicious entities cannot disrupt the calculations or steal sensitive information.

FedGMMAT currently relies on a round-robin (or cyclic) schedule, which requires sites to wait for their turns without asynchronous updates. In addition, the current implementation relies on TenSEAL framework which may not be optimal for the communication bandwidth. Thus, our future work will focus on refining the current method by adding secure multi-computation protections, embedding a tree-based asynchronous communication framework, and using lightweight communication protocols.

Finally, the statistical model that GMMAT uses models the random effects as random intercepts. The models can be extended to accommodate random slopes models to account for the structured heterogeneity of the genetic effects coefficients. This, however, would require substantial changes to the current algorithm. This model should also be tuned well to ensure that null model is not overly relaxed or stringent, which may diminish the model's power to detect associations.

## 4 Methods

### 4.1 Problem setup

We focus on the single-variant test for binary traits under a federated setting with numerous sites (also referred to as nodes or clients). The trait's occurrence probability is modeled by following logistic mixed-effects model:

$$g(\pi_{ij}) = \mathbf{X}_{ij}\boldsymbol{\alpha} + G_{ij}\beta + b_{ij}$$

where $g(\cdot)$ is the logit link function (i.e., $logit(x) = \log\left(\frac{x}{1-x}\right)$), and $\pi_{ij} = P(y_{ij} = 1|\mathbf{X}_{ij}, G_{ij}, b_{ij})$ is the probability of the binary trait (i.e., disease status) for subject $i$ in the site $j$, and $y_{ij}$ denotes the binary trait status of individual $i$ on site $j$ (Table 7). Subscripts $i < n_j$ and $j < S$ denote the specific subject $i$ on site $j$, and $n_j$ denotes the number of subjects on site $j$.

$\mathbf{X}_{ij}$ is a $p$-dim vector of fixed covariates (e.g., population principal components, gender), $G_{ij}$ is the genotype, a scalar value indicating the dosage of alternate allele at the genetic variant of interest. When referring to the local data on site $j$, we use a single subscript, e.g., $\mathbf{X}_j$, $\mathbf{G}_j$ and $\mathbf{b}_j$. On the site $j$, $\mathbf{X}_j^\top = (\mathbf{X}_{1j}^\top, \mathbf{X}_{2j}^\top, \ldots, \mathbf{X}_{n_j j}^\top)$ is a $n_j \times p$ matrix and $\mathbf{G}_j = (G_{1j}, G_{2j}, \ldots, G_{n_j}j)^\top$ is a $n_j$-dim vector (for a single variant of interest), and $\mathbf{b}_j = (b_{1j}, b_{2j}, \ldots, b_{n_j j})^\top$ is an $n_j$-dim vector, which quantifies the random polygenic effects due to cryptic (i.e., unknown) kinship among subjects, and different experimental designs (e.g., longitudinal studies. This vector is assumed

**Table 7. Nomenclature table.**

| | | | |
|---|---|---|---|
| $i$ | Index of samples | $\boldsymbol{\alpha}$ | Parameters of covariate |
| $j$ | Index of federated node | $G_j$ | $n_j \times k$ genotype matrix of node $j$ |
| $J$ | Total number of federated nodes | $\mathbf{V}$ | $n \times n$ kinship matrix |
| $p$ | Number of covariates | $b_{ij}$ | Random effect of $i$-th sample in $j$-th node |
| $n$ | Total number of samples | $\Sigma$ | $n \times n$ matrix |
| $n_j$ | Number of samples in node $j$ | $\mathbf{b}_j$ | Vector of random effect of $j$-th node |
| $\pi_{ij}$ | Probability of $i$-th sample in $j$-th node | $y_{ij}$ | The outcome of sample $i$ from node $j$ |
| $g$ $(\cdot)$ | Logit link function | $\mathbf{X}_{ij}$ | $p$-dim vector of covariates of sample $i$ from node $j$ |
| $G_{ij}$ | The genotype of $i$-th sample in $j$-th node | $X_j$ | $n_j \times p$ covariate matrix of node $j$ |
| $\beta$ | Parameters of genotype effect | $\mathbf{W}$ | $n \times n$ diagonal weight matrix |
| $\tau$ | Variance component parameter for random effects | $\Sigma_{(j)}^{-1}$ | $n \times n_j$ matrix of vertically split part of $\Sigma^{-1}$ in node $j$ |
| $\Omega$ | $n \times p$ matrix, product of $\Sigma^{-1}\mathbf{X}$ in node $j$ | $\mathbf{P}$ | $n \times n$ projection matrix |
| $\Omega_j$ | $n \times p$ matrix, product of $\Sigma_{(j)}^{-1}\mathbf{X}_j$ in node $j$ | $\Omega_{(j)}$ | $n_j \times p$ matrix of horizontally split part of $\Sigma^{-1}\mathbf{X}$ in node $j$ |
| $t$ | Index of iteration | $k$ | number of SNPs |

to be sampled from a zero mean multivariate distribution whose covariance matrix is proportional to the relatedness (i.e., design or kinship) matrix, $V$.

**Parameters.** $\boldsymbol{\alpha}$ is a $p$-dim vector of the fixed covariate effect sizes, and $\beta$ is the alternate allele dosage effect size (i.e., genotype). We also assume $\boldsymbol{b} \sim \mathcal{N}(0, \tau\mathbf{V})$ is a $n$-dim vector where $\boldsymbol{b}^\top = (\mathbf{b}_1^\top, \mathbf{b}_2^\top, \dots, \mathbf{b}_J^\top)$, $\tau$ is the variance component parameter for random effects, and $V$ is an $n \times n$ matrix refers to kinship relationship matrix for all samples across all sites, which can be calculated using secure kinship estimation methods [32]. We denote the variance of trait as $v$ $(\cdot)$, where $v(\pi_{ij}) = Var(y_{ij}|\boldsymbol{b}) = \pi_{ij}(1 - \pi_{ij})$ for the binary traits.

FedGMMAT implements the score test that is formulated by GMMAT algorithm [46] in a federated setting. This procedure consists of two main steps. First step is the null model fitting, which uses only the covariate information without exchanging any sensitive genotype information among sites. The information exchanges are protected by high level data aggregation, and encryption among sites. Second step is the variant scoring step, which evaluates the significance of observed genotype effect sizes given the null model. The separation of the two steps improves efficiency of the overall process. This approach is advantageous from privacy-preservation aspect because it flexibly separates the model fitting and scoring steps, each of which can be federated independently. Depending on the goal of the analysis, sites can make use of the flexibility of GMMAT's approach to provide efficiency while privacy can be preserved under local regulations of the sites. We describe below the details of the model fitting protocols (S4 and S5 Figs).

## 4.2 Setting up the network

We assume that the sites have secure channels among each other and with the Central Server. The Central Server is assumed to be a trusted entity (e.g., NIH, COLLAGENE server [50], etc) who does not deviate from protocols and the main risk is hacking or accidental leakage of datasets from the Central Server. It is therefore necessary to minimize unencrypted data transfers to the server. We recommend using AWS cloud-based operations where sites can setup shared volume mounts to exchange data with very low latency. This can be programmatically setup among the sites. Alternatively, the sites can exchange datasets using scp-based protocols [50].

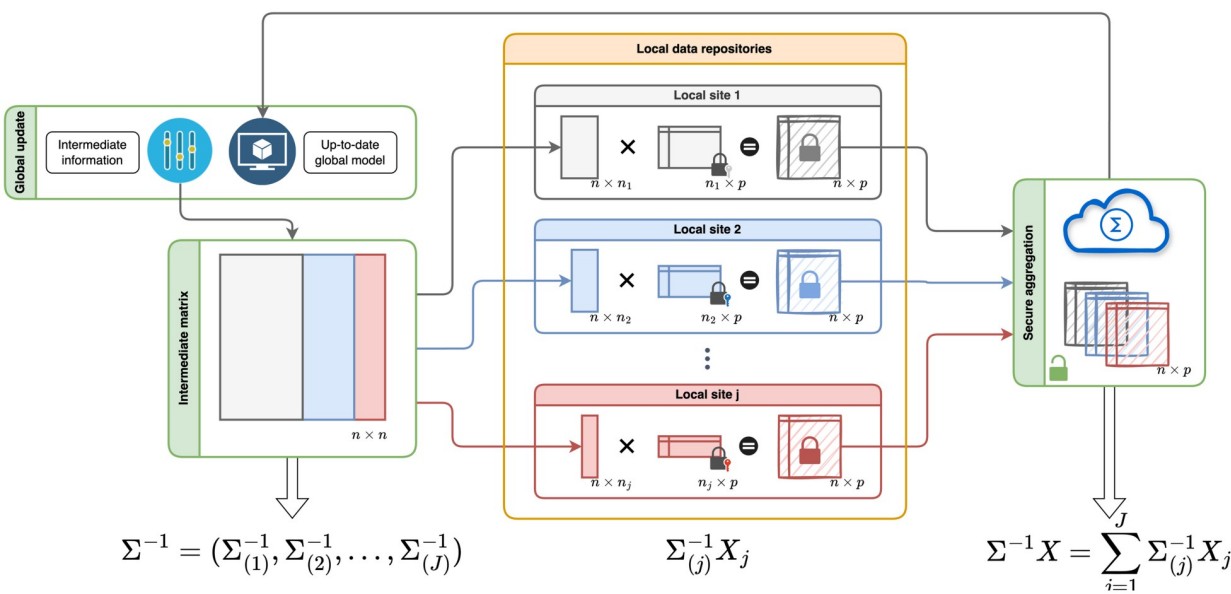

**Fig 6. Matrix Splitting at the Central Server and aggregation among local data repositories.** In the FedGMMAT framework, each local data repository will maintain its unique dataset locally and gather intermediate model information from Global updates. The vertical splitting of $\Sigma^{-1}$ among local repositories is illustrated in the figure as an example. Each site receives $n_j \times n$ sized covariate matrices ($X_j$), which are multiplied locally with $n_j \times p$ sized matrices. The resulting $n \times p$ matrices are encrypted and aggregated among sites via round-robin schedule and sent to the server. Splitting and aggregation of other matrices is accomplished with similar protocol.

## 4.3 Secure aggregation and partitioning of intermediate matrices

FedGMMAT's protocols make extensive use of the *encrypt-aggregate-decrypt-partition* procedure, depicted in Fig 6 and S2 Fig. Most operations in FedGMMAT require the sites to aggregate intermediate matrices that are of the same size. We describe the details of these procedures:

By default, FedGMMAT uses Homomorphic Encryption-based encryption among the sites. This is initiated by the Central Server by generating a key pair and broadcasting the public and homomorphic evaluation keys among the sites (i.e., Galois, Relinearization). Encryption and summation at sites, and decryption at the Central Server uses TenSEAL library's CKKS scheme.

In some parts, HE-based calculations are computationally heavy since FedGMMAT's current framework relies only on the aggregations via summation (e.g., score testing stage). To decrease the computational and storage requirements, Central Server can use an alternative encryption methodology that uses a one-time-pad-like (OTP-like) strategy where Central Server shares noise matrices with large variance that masks and intermediate data. In a nutshell, whenever an intermediate $n \times p$ matrix $\boldsymbol{\Sigma}^{-1}_{(j)}\boldsymbol{X}_j$ matrix (from the original protocol) is to be encrypted and aggregated among sites, the server provides a noise matrix, denoted $N_j^{(server)}$, of size $n \times p$ to site $j$. This matrix represents the symmetric encryption key for the respective site that is shared by means of a secure and secret pseudo-random seed for generating the random matrix. The encryption of $\boldsymbol{\Sigma}^{-1}_{(j)}\boldsymbol{X}_j$ is simply addition of the noise matrix:

$$enc(\boldsymbol{\Sigma}^{-1}_{(j)}\boldsymbol{X}_j) = \boldsymbol{\Sigma}^{-1}_{(j)}\boldsymbol{X}_j + N_j^{(server)}$$

In this equation, we can simply add the noisy (encrypted) matrices from all sites:

$$\sum_j enc(\mathbf{\Sigma}_{(j)}^{-1}\mathbf{X}_j) = \mathbf{\Sigma}^{-1}\mathbf{X} + \sum_j N_j^{(server)}$$

which is sent to the Central Server. After this step, the Central Server can "decrypt" the aggregated sum (first terms of right hand side) by simply removing the encryption keys (i.e., the noise terms), to obtain $\mathbf{\Sigma}^{-1}\mathbf{X}$. This is, however, not secure because the Central Server can gain access to the original matrix $\mathbf{X}$ since server already knows $\mathbf{\Sigma}^{-1}$. To circumvent this issue, the sites make use of a partitioned noise matrix, denoted by $N_{(j)}^{(part)}$, before encryption (S2 Fig). At site $j$, a noise matrix is sampled such that it contains entries only at the $j$th site's partition on the $n \times p$ matrix. This matrix is known only at site $j$ and protects site-$j$'s partition. For site $j$, partitioned noise matrix is an $n \times p$ matrix but it has non-zero entries only in the $n_j$ rows (S2 and S3 Figs). This way, when we sum the partitioned noise matrices from all sites, we obtain a full noise matrix, i.e., $\sum_j N_{(j)}^{(part)}$ is a dense noise matrix whose partitions are known only to the sites and no other entity.

Thus, before encryption (via OTP-like noise addition), each site adds their share of the partitioned noise matrix. After aggregation, the Central Server receives:

$$\sum_j enc(\mathbf{\Sigma}_{(j)}^{-1}\mathbf{X}_j + N_{(j)}^{(part)}) = \mathbf{\Sigma}^{-1}\mathbf{X} +$$

$$\sum_j N_{(j)}^{(part)} + \sum_j N_j^{(server)}$$

The central server "decrypts" the aggregated matrix by removing $\sum_j N_j^{(server)}$, and obtains the aggregated data with dense noise matrix:

$$dec(\sum_j enc(\mathbf{\Sigma}_{(j)}^{-1}\mathbf{X}_j + N_{(j)}^{(part)})) =$$

$$\mathbf{\Sigma}^{-1}\mathbf{X} + \sum_j N_{(j)}^{(part)}$$

Note that Central Server cannot further remove partitioned noise matrices since these are only known to the sites. The Central Server finally partitions the noise aggregated matrix according to each site's row partitioning. For site-$k$, Central Server partitions the noisy matrix as:

$$(\mathbf{\Sigma}^{-1}\mathbf{X} + \sum_j N_{(j)}^{(part)})_{(k)} = (\mathbf{\Sigma}^{-1}\mathbf{X})_{(k)} + (\sum_j N_{(j)}^{(part)})_{(k)}$$

This partition is sent to site-$k$. By definition, the partitioning of the total noise matrix in above equation is the partitioned noise matrix at site $k$:

$$(\sum_j N_{(j)}^{(part)})_{(k)} = N_{(k)}^{(part)}[i : i + n_k, :]$$

where $N_{(k)}^{(part)}[i : i + n_k, :]$, denotes the rows of partitioned noise matrix of site-$k$ that is known only to site-$k$. Thus, site-$k$ can remove this noise component and obtain $(\mathbf{\Sigma}^{-1}\mathbf{X})_{(k)}$, which completes the protocol.

While describing the protocols, we may skip the explicit description of the partitioned noise matrices for brevity (See S4 and S5 Figs for a more complete description of the full details of FedGMMAT protocols with noise matrices). Also, it should be noted that the OTP-like encryption mechanism is used as an efficient alternative to HE-based approach. In the above

aggregation protocol, "*enc*" and "*dec*" operations can be replaced by HE encryption and decryption mechanisms and the partitioned noise mechanism can be used as presented above without any modifications. Clearly, HE operations does not need the server noise matrices, which are OTP-like encryption specific noise matrices.

By default, FedGMMAT uses a round-robin schedule to aggregate matrices, which are then sent to the Central Server. Of note, this schedule is not optimal but we stick to this schedule in below description for simplicity of presentation and implementations. A more optimal schedule can be implemented by sending all matrices to one site, aggregating simultaneously and sending to the main server.

## 4.4 Fixed effect size $\alpha_0$ initialization

In order to test the significance of the genotype effect, we need to fit a federated logistic mixed model under null hypothesis $H_0 : \beta = 0$, the null model for trait probability of subject $i$ on site $j$ is

$$g(\tilde{\pi}_{ij}) = \mathbf{X}_{ij}\boldsymbol{\alpha} + b_{ij}$$

The likelihood function for this full model is not tractable as it has no closed form solution and we adopt the method used in Breslow et al. [59]. To solve the initial estimates of fixed effect parameters we use the Laplace approximation of the integral. We then derive closed-form solutions of fixed-effect coefficients $\hat{\boldsymbol{\alpha}}$ and $\hat{\boldsymbol{b}}$ in a federated way in mixed effect model fitting step. First, we start from obtaining the initial estimate $\hat{\boldsymbol{\alpha}}$ without random effects using Laplace approximation. The log-likelihood of the initial estimate of $\boldsymbol{\alpha}$ under $\beta = 0$ and $\tau = 0$ is of the form

$$\tilde{\ell}(\boldsymbol{\alpha}) = \sum_{j=1}^{J} \sum_{i=1}^{n_j} (y_{ij} \log(\pi_{ij}) + (1 - y_{ij})\log(1 - \pi_{ij})).$$

The optimization of such a log-likelihood can be split among sites; where each site locally calculates its gradient and hessian, encrypts them, and sites aggregate the matrices in the round-robin scheme (Fig 6). These matrices can be recovered as following:

**Gradient.**

$$\frac{\partial \tilde{\ell}(\boldsymbol{\alpha})}{\partial \boldsymbol{\alpha}} = \sum_{j=1}^{J} \frac{\partial \tilde{\ell}_j(\boldsymbol{\alpha})}{\partial \boldsymbol{\alpha}}$$

where $\tilde{\ell}_j = \sum_{i=1}^{n_j} (y_{ij} \log(\pi_{ij}) + (1 - y_{ij})\log(1 - \pi_{ij}))$

**Hessian.**

$$\mathbf{H} = \frac{\partial^2 \tilde{\ell}(\boldsymbol{\alpha})}{\partial \boldsymbol{\alpha}^2} = \sum_{j=1}^{J} \frac{\partial^2 \tilde{\ell}_j(\boldsymbol{\alpha})}{\partial \boldsymbol{\alpha}^2} = \sum_{j=1}^{J} \mathbf{H}_j$$

The log-likelihood of the null model can be optimized under federated learning settings by Newton's method:

$$\boldsymbol{\alpha}^{(n+1)} = \boldsymbol{\alpha}^{(n)} - \tilde{\ell}'/\mathbf{H}$$

Of note, this is a logistic regression fitting loop. The fitting loop runs until convergence to initialize covariates' fixed effects vector $\tilde{\boldsymbol{\alpha}}_0$ in the null model. This vector is shared among sites and is used as the initial fixed-effect estimate of the covariates. The Hessian and gradients are

encrypted at each iteration and aggregated using round-robin schedule. $\alpha$ is updated at the Central Server, which gets access to the plaintext Hessian matrix ($p \times p$) and the gradient vector ($p$ long).

## 4.5 Mixed effect size estimation

Site $j$ uses the current null model parameters and calculates a local weight vector $\mathbf{w_j} = [w_{1j}, w_{2j}, \ldots]$ such that $\mathbf{w}_{ij} = 1/(v(\pi_{ij})[g'(\pi_{ij})]^2) \times \mathbf{1}_{n_j}$, where $g'(\pi_{ij}) = 1/v(\pi_{ij})$. This is a highly aggregated function of the current null model parameters and it relies only on the covariate information ($X_{ij}$) (no reliance on genotype). Each site calculates $\mathbf{w_j}$ and sends it independently to the central server for construction of a global weight matrix $\mathbf{W} = diag(\mathbf{w}_1^\top, \mathbf{w}_2^\top, \ldots, \mathbf{w}_J^\top)$. Let $\boldsymbol{\Sigma} = \mathbf{W}^{-1} + \tau \mathbf{V}$ and split it vertically with the sample size of each node, as form $\boldsymbol{\Sigma}^{-1} = (\boldsymbol{\Sigma}_{(1)}^{-1}, \boldsymbol{\Sigma}_{(2)}^{-1}, \ldots, \boldsymbol{\Sigma}_{(J)}^{-1})$ (Fig 5). Central server sends the back these to each site. Note that this is the most intensive step in network transfers. Central server also sends the partitions of $(V \boldsymbol{\Sigma}^{-1})_{(j)}$ to each site. Note that these are sent to individual sites without encrypting since they are not broadcast. In addition, these matrices can be submitted by quantizing to 32-bit floating point representations to decrease storage and network bandwidth. The network and computational requirements can be further decreased substantially if sites can assume there are no cross-site relatedness between sites, in which case $W$ is a block diagonal matrix and each site can form this matrix locally without input from central server. We believe this simplification is reasonable since most relatedness will be confined to within sites and will be reasonably small across different sites. This can be evaluated after calculation of the kinship matrix and deciding whether certain cross-site relatedness should be considered.

Each site $j \in \{1, \ldots, S\}$ calculates $\boldsymbol{\Omega}_j = \boldsymbol{\Sigma}_{(j)}^{-1} \mathbf{X}_{(j)}$, ($n_j \times p$ matrix). These are next securely aggregated using round-robin schedule (Figs 5 and 6, and S4 Fig; Step 4) among sites to compute $\boldsymbol{\Omega} = \sum_{j=1}^J \boldsymbol{\Omega}_j$. Next, central server horizontally partitions $\Omega$ into $\Omega_{(1)}, \Omega_{(2)}, \ldots, \Omega_{(J)}$ and sends them to corresponding sites.

Each site calculates $\boldsymbol{\Xi}_j = \mathbf{X}_j^\top \boldsymbol{\Omega}_{(j)}$ ($p \times p$ matrix, S4 Fig; Step 5). This matrix is encrypted, and aggregated among sites to calculate $\Xi$. Aggregated matrix $\Xi$ is sent to central server. Each site updates their random effect estimate (S4 Fig; Step 6):

$$\hat{\boldsymbol{b}}_j = \tau(\mathbf{V}\boldsymbol{\Sigma}^{-1})_{(j)}(\tilde{\mathbf{Y}}_j - \boldsymbol{\Psi}_j)$$

In the above, $\boldsymbol{\Psi}_j$ and $\tilde{\mathbf{Y}}_j$ are locally calculated at each site as following:

$$\begin{aligned}\tilde{\mathbf{Y}}_j &= g(\boldsymbol{\pi}_j) + g'(\boldsymbol{\pi}_j)(\boldsymbol{y}_j - \boldsymbol{\pi}_j) \\ \boldsymbol{\Psi}_j &= \mathbf{X}_j\hat{\boldsymbol{\alpha}}\end{aligned}$$

These are securely aggregated among sites and sent to central server. Central server decrypts and broadcasts $\hat{\boldsymbol{b}}$ to be used in the next iteration (S4 Fig; Step 6). Sites also send $\tilde{\mathbf{Y}}_j$, which are concatenated at the central server to generate $\tilde{\mathbf{Y}}$ ($n \times 1$) vector. Next, central server calculates the updated covariate information:

$$\hat{\boldsymbol{\alpha}} = \boldsymbol{\Xi}^{-1} \left( \sum_{j=1}^J \boldsymbol{\Omega}_j \right)^\top \tilde{\mathbf{Y}}$$

To estimate this, sites encrypt-aggregate-decrypt-partition $\{\boldsymbol{\Omega}_{(j)}^\top \tilde{\mathbf{Y}}_j\}$ then multiply on the left with $\Xi$ to obtain $\hat{\boldsymbol{\alpha}}$, the new estimate for the fixed covariate effects (S4 Fig, Step 7).

At this stage, all sites have a new estimate for random effects vector and the fixed covariate effects of the null model.

**Variance component updates.**   The final step of calculation is the update of variance component $\tau$. We ignore the dependence of $\mathbf{W}$ on $\tau$ then adopt Pearson Chi-squared approximation (Supplementary Information Equation 2) to the deviance.

At this step, it is necessary for the server to calculate the projection matrix $P = \mathbf{\Sigma}^{-1} - \Omega\Xi\Omega^\top$ that is of size $n \times n$ (S5 Fig, Protocol for Calculating P). At the step, sites use a random rotation of $\Omega$ via a common random $p \times p$ matrix $Q$ sampled from $N(0, n)$. All sites $j < J$ synchronously send $\Omega_j Q$ ($n_j \times p$) and $(Q^{-1} \, \mathbf{\Xi} \, \mathbf{\Omega}_j^\top)$ ($p \times n_j$). These matrices are sent directly to the central server and are not encrypted further to protect against other sites. These matrices are protected by the random rotations (known only to the sites, not the central server) on the right and left and does not leak immediate identifying information in case of hacking or accidental leakages from the Central Server. Central Server calculates $P$:

$$P = \mathbf{\Sigma}^{-1} - \left( \sum_j \mathbf{\Omega}_j \, Q \right) \left( \sum_j Q^{-1} \, \mathbf{\Xi} \, \mathbf{\Omega}_j^\top \right)$$
$$= \mathbf{\Sigma}^{-1} - (\mathbf{\Omega} \, \mathbf{\Xi} \, \mathbf{\Omega}^\top)$$

After obtaining We optimize the Quasi-Likelihood objective function with respect to variance component parameter $\tau$ by Average Information REML algorithm [60]. This step is accomplished at the Central Server using $\mathbf{V}$, $\mathbf{P}$, and $\tilde{Y}$ (S4 Fig, Step 8). The central server broadcasts the scalar $\tau$ to all sites. At this stage, the sites have access to updated set of fixed and random effects parameters.

## 4.6 Federated score test and assignment of p-values

After null model parameter estimation is completed, the sites use the null model parameters with genotype data to assign P-values that determines the level to deem the variants as significantly associated with the trait.

For each of sites $j < J$, the Central Server sends the $n_j \times n$ horizontal partitions of $P$ denoted by $\mathbf{P}_{(j)}$ to site $j$. This is needed to be performed only once before score testing of variants. Central Server also calculates $n_j \times 1$ vector $(\mathbf{P}\tilde{Y})_{(j)}$ to site $j$ (S5 Fig; Protocol for $P$ and Score Calculation Protocol).

For each chunk of size $k$, site-$j$ calculates $G_j^\top (\mathbf{P}\tilde{Y})_{(j)}$ ($k \times 1$ vector), encrypt and aggregated to calculate $G^\top \mathbf{P}\tilde{Y}$. This matrix is decrypted at the Central Server and broadcast to all sites (S5 Fig; Score Calculation protocol).

The variance of the score test is $diag(\mathbf{G}^\top \mathbf{P}\mathbf{G})$. In order to reconstruct the variance FedGMMAT uses the following steps (S5 Fig, Protocol for Calculating Score Variance): 1) Calculate $\mathbf{G}^\top \mathbf{P}$: $\mathbf{G}^\top \mathbf{P}$ can be divided into site-level $\sum_{j=1}^{J} \mathbf{G}_j^\top \mathbf{P}_j$; then 2) horizontally partitioned $\mathbf{G}^\top \mathbf{P}$ with respect to sample size, and distribute corresponding part $\mathbf{G}^\top \mathbf{P}_{(j)}$ to clients and calculate the variance matrix $\mathbf{G}^\top \mathbf{P}\mathbf{G} = \sum_{j=1}^{J} \mathbf{G}^\top \mathbf{P}_{(j)}\mathbf{G}_j$.

This concludes the score statistic and variance estimation protocol for the $k$-variant chunk. The p-values are assigned using the asymptotic approximation of the null distribution as the Chi-squared distribution with 1 degree-of-freedom calculated using the the score statistic divided by its variance.

### 4.7 Federated learning workflow and communications

The workflow of the fedGMMAT can be summarized in the following four major steps:

1. **Key Generation and Setup**: The central server generates an asymmetric public-private key pair and shares the public keys with all sites. This step and all homomorphically encrypted operations are implemented using TenSEAL package [47]. When using One-Time-Pad-like encryption, the server initiates the random number generator to generate the noise matrix seed values to be used to generate the random matrices (pads) for encryption.

2. **Project initiation**: The central server receives the training requests from clients. During this process, model metadata will be collected from each client, including the number of clients in the federated project $J$, the sample size of each client $n_j$, and the number of covariates.

3. **Null model fitting**: In this step, the federated clients will jointly train the null model, a logistic regression model, by following steps:

   (a) **Broadcast model parameters**: The central server initiates the process of null model learning by broadcasting the parameters and public key to clients.

   (b) **Aggregate local information**: Each federated client $j$ will calculate and pass the homomorphic encrypted local information (i.e. Gradient $\tilde{\ell}_j$ and Hessian matrix $\mathbf{H}_j$) with secure aggregation framework in Fig 7.

   (c) **Update model parameters**: The central server will retrieve the encrypted aggregated model information from the last client and decrypt it with the private key (Fig 7). Then update the model parameters with the aggregated information.

   (d) **Broadcast model status**: The central server will first broadcast the up-to-date global parameters to all clients. Then, it will determine whether the model is converged or not by the log-likelihood score. If the difference from the last iteration is within $1 \times 10^{-6}$, then stop the iteration. If not, goes to step a).

4. **Mixed-effects model fitting**: In this step, the sites collaborate on mixed-effect model training. Initializing from the null model, the federated learning model will estimate the fixed-effect coefficients, mixed-effect coefficients, and mixed-effect hyper-parameters with the following steps:

   (a) **Broadcast initial model parameters**: The central server will set the parameters from the null model as the initial fix-effect coefficients $\tilde{\boldsymbol{\alpha}}_0$, and the variance of targets as the initial mixed-effect hyper-parameters $\tilde{\tau}_0$, and HE public key. Then send them to the federated clients.

   (b) **Aggregate local information**: Once the federated client $j$ receives the parameters, it will compute the local information $\boldsymbol{\Xi}_j, \boldsymbol{\Omega}_j, \tilde{\mathbf{Y}}_j, \boldsymbol{\Psi}_j$ as defined previously. Then cyclically aggregate and send the encrypted aggregated information $enc(\boldsymbol{\Xi}), enc(\boldsymbol{\Omega}), enc(\hat{\boldsymbol{b}})$ to the central server (Fig 7).

   (c) **Update model parameters**: The central server decrypts the aggregated information then updates variance component estimate $\hat{\tau}$, fixed-effect coefficient $\hat{\boldsymbol{\alpha}}$, and random effects coefficient $\hat{\boldsymbol{b}}$.

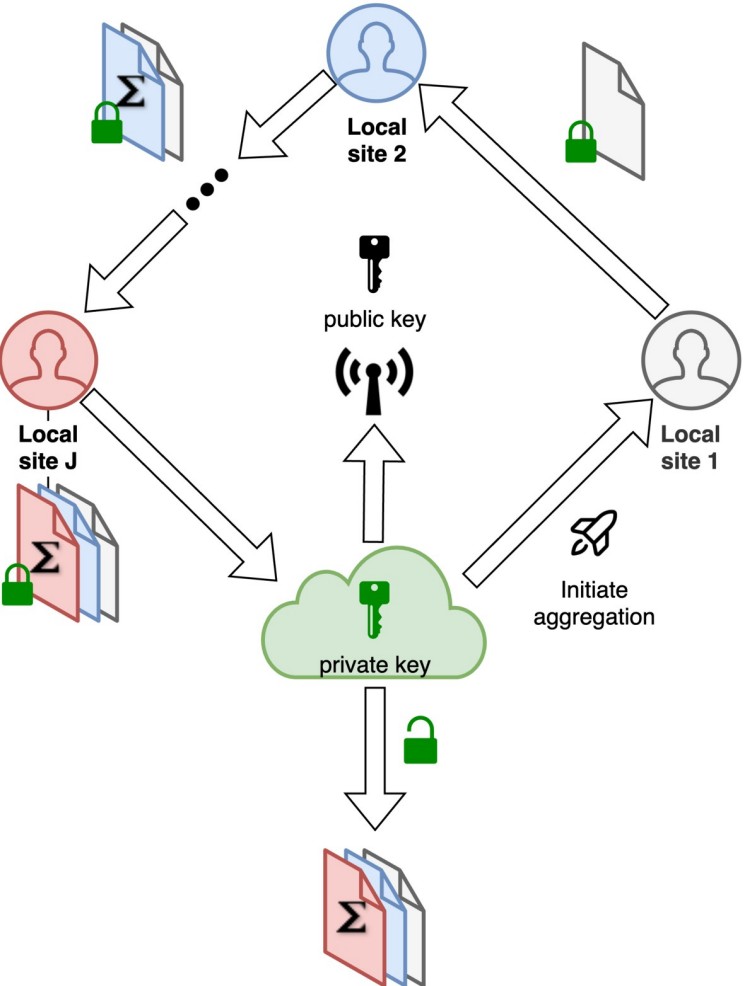

**Fig 7. Key setup and aggregation via round-robin schedule.** The central server initiates the public/private key pair at the initialization and broadcasts the public key to all sites. Aggregation of split matrices are performed among sites by taking turns in aggregation. Each site encrypts their matrix "share" before aggregation. Symmetric keys (secret seeds) for OTP-like encryption is sent to all sites at the scoring stage whenever an aggregation is performed (S2 and S3 Figs).

 (d) **Broadcast model status**: After updating and broadcasting the up-to-date model parameters, the central server will determine whether the model is converged with the difference of parameters from the last iteration. If it has not converged, go to step b).

5. **Federated learning on score test**: Once the parameter space $(\hat{\boldsymbol{\alpha}}, \hat{\tau})$ is optimized under the objective function (Equation S1.3 (1)), the FedGMMAT can compute the score test. First, the central server will send the corresponding local model residual $(\mathbf{P}\tilde{\mathbf{Y}})_{(j)}$ to each federated node. Then the score test statistics $T$ can be collected and aggregated from federated nodes by protocol S5 Fig. The final step is to calculate variance $GPG^{\top}$ with a developed federated learning protocol (S5 Fig; Protocol for Calculating Variance).

The communicated information is summarized in Table 8, and algorithms 1, 2, 3. These are also depicted in S2 and S3 Figs.

**Table 8. Communication information.**

| | Client to Server | Data size | Server to Client | Data size |
|---|---|---|---|---|
| Null model | $enc(\sum_{j=1}^{J} \tilde{\ell}_j^{(t)})$ | $p$-dim vector | $\boldsymbol{\alpha}^{(t)}$ | $p$-dim vector |
| | $enc(\sum_{j=1}^{J} \mathbf{H}_j^{(t)})$ | $p \times p$ matrix | $pk$ | scalar |
| Mixed model | | | $(\mathbf{V}\boldsymbol{\Sigma}^{-1})_{(j)}$ | $n \times n_j$ matrix |
| | $enc(\sum_{j=1}^{J} \boldsymbol{\Xi}_j)$ | $p \times p$ matrix | $\boldsymbol{\alpha}^{(t)}$ | $p$-dim vector |
| | $enc(\sum_{j=1}^{J} \boldsymbol{\Omega}_j)$ | $n \times p$ matrix | $\boldsymbol{b}^{(t)}$ | $n$-dim vector |
| | $enc(\hat{\boldsymbol{b}}_j)$ | $n$-dim vector | $\hat{\tau}$ | scalar |
| | | | $pk$ | scalar |
| Score test | $enc(\mathbf{G}_j^\top (\mathbf{P}\tilde{\mathbf{Y}})_{(j)})$ | $nSNPs$-dim vector | $enc((\mathbf{P}\tilde{\mathbf{Y}})_{(j)})$ | $n_j$-dim vector |
| | | | $pk$ | scalar |

**Algorithm 1** Federated Initial Fixed-effects Estimate

```
Null model initialization;
Central server:
Broadcast initial global model parameters α⁽⁰⁾, public key pk;
for t ∈ [0, 1, 2, ..., T] do
  for j ∈ [1, 2, ..., J] do
    Federated trainer j:
    1. Receive the up-to-date global model parameters α⁽ᵗ⁾ and the HE
       public key pk from the Central server;
    2. Update the local gradient ℓ̃ⱼ⁽ᵗ⁾ and Hessian matrix Hⱼ⁽ᵗ⁾, and encrypt
       it with pk;
    3. Aggregate enc(ℓ̃ⱼ⁽ᵗ⁾) and enc(Hⱼ⁽ᵗ⁾) and pass the information (Fig 7);
  end
  Central server:
  1. Receive the encrypted HE information enc(∑ⱼ₌₁ᴶ ℓ̃ⱼ⁽ᵗ⁾) and enc(∑ⱼ₌₁ᴶ Hⱼ⁽ᵗ⁾)
     (Fig 7);
  2. Decrypt and reveal the aggregated information ∑ⱼ₌₁ᴶ ℓ̃ⱼ⁽ᵗ⁾ and ∑ⱼ₌₁ᴶ Hⱼ⁽ᵗ⁾
     with private key;
  3. Update and broadcast the model parameter with α⁽ᵗ⁺¹⁾ = α⁽ᵗ⁾ − ℓ̃′/H;
  4. if Δ(α) < 1 × 10⁻⁶ then
     α̃₀ = α⁽ᵗ⁾ and stop.
  end
end
```

**Algorithm 2**: Federated Mixed-effects Model

```
Mixed model initialization;
Central server:
Broadcast initial fixed-effect coefficient α̃₀, initial mixed-effects
parameter τ̃₀, and public key pk;
for t ∈ [0, 1, 2, ..., T] do
  for j ∈ [1, 2, ..., J] do
    Federated trainer j:
    1. Receive the up-to-date global model parameters α̂⁽ᵗ⁾, τ̂⁽ᵗ⁾, Σ⁻¹₍ⱼ₎,
       (VΣ⁻¹)₍ⱼ₎, and the HE public key pk from the Central server;
    2. Update the local information Ξⱼ, Ωⱼ, Ỹⱼ, Ψⱼ, then compute the local
       bias estimator b̂ⱼ and encrypt it with pk;
    3. Aggregate enc(Ξⱼ), enc(Ωⱼ), enc(b̂ⱼ) and pass the information (Fig
       7);
  end
```

**Central server:**

1. Receive the encrypted HE information $enc(\mathbf{\Xi})$, $enc(\mathbf{\Omega})$, $enc(\hat{\boldsymbol{b}})$ (Fig 7);
2. Decrypt and reveal the aggregated information $\sum_{j=1}^{J} \mathbf{\Xi}_j$, $\sum_{j=1}^{J} \mathbf{\Omega}_j$, $\sum_{j=1}^{J} \hat{\boldsymbol{b}}_j$ with private key;
3. Update and broadcast the model parameter $\hat{\boldsymbol{\alpha}}$ and $\hat{\boldsymbol{b}}$;
4. **if** $\Delta(\hat{\boldsymbol{\alpha}})$ & $\Delta(\hat{\boldsymbol{b}}) < 1 \times 10^{-6}$ **then**

$\hat{\boldsymbol{\alpha}} = \hat{\boldsymbol{\alpha}}^{(t)}$, $\hat{\boldsymbol{b}} = \hat{\boldsymbol{b}}^{(t)}$ and stop.

**end**

**end**

**Algorithm 3**: Federated Score Test

Projection matrix **P** initialization;

**Central server:**

Split **P** and send the corresponding part $\mathbf{P}_j$ to client $j$.

When using HE: Broadcast the public key $pk$;

When using OTP-like encryption, initialize the seed generator for generating encryption keys for OTP-like encryption, i.e., $N_j^{(server)}$ matrices;

**for** $j \in [1, 2, ..., J]$ **do**

 **Federated trainer** $j$:

 1. Receive the local model residual $(\mathbf{P}\tilde{\mathbf{Y}})_{(j)}$ from the central server;
 2. Encrypt-aggregate the score test statistics $T = \sum_{j=1}^{J}(\mathbf{G}_j^{\top}(\mathbf{P}\tilde{\mathbf{Y}})_{(j)})$;
 3. Encrypt-aggregate-decrypt partition $\mathbf{G}_j^{\top}\mathbf{P}_{(j)}$ and share $(\mathbf{G}^{\top}\mathbf{P})_{(j)}$ matrix to sites.;
 4. Encrypt-aggregate-decrypt to calculate $\sum_j (\mathbf{G}^{\top}\mathbf{P})_{(j)}\mathbf{G}_j = \mathbf{G}^{\top}\mathbf{PG}$, the score variance for the current chunk of $k$ variants.

**end**

**Central server:**

1. Receive the encrypted information $enc(T)$ (Fig 7);
2. Decrypt and reveal the aggregated information $T$ with private key;
3. Broadcast the score test statistic $T$ among federated trainers.

## 5 Genotype and phenotype simulations

The simulated datasets in the experiments are generated using the 1000 Genomes Project's phase 3 data. For the simulated datasets referred to as homogeneous and heterogeneous populations, we simulated 400 pedigrees of 16 members (8 founders and 8 descendants) comprising 6400 individuals [49]. Simulation of each descendant is performed by assigning the alleles on each haplotypes for each variant using the parent's variants. We randomly selected 6000 individuals to be split among the 3 sites.

Homogeneous cohort is simulated by selecting the 8 founders in each family from the Great Britain population in the 1000 Genomes Project data. In order to have enough founder subjects, the founder GBR population was resampled to generate 3200 founder individuals in 400 pedigrees (8 founder in each population).

Heterogeneous cohort is simulated by selecting the 8 founders in each family randomly from one of Great British (European), Mexican in Los Angeles (American), or Yoruban from Ibadan Nigeria (African) populations. Each population was similarly resampled to be used as distinct founder subjects for the 800 families.

For simulated the phenotypes, we first randomly selected 20 causal variants to be used for simulating the genetic component of phenotypes. Each variant's affect is selected to account for approximately half of the phenotypic variance. Gender effects are modeled with fixed 0.01 and environmental noise standard deviation is set to 0.5. Each phenotype was simulated using

logit link function as:

$$
\begin{aligned}
Y_i &\lesseqgtr logit(\mathbf{G_i}\beta_\mathbf{i} + X_i \cdot 0.01 + \epsilon_i) \\
\epsilon_i &\sim N(0, 0.5)
\end{aligned}
$$

## Supporting information

**S1 Text. The supplementary text.**
(PDF)

**S1 Fig. QQ-plot for the 6500 variants identified using the dataset with kinship heterogeneity between case and control cohort.** (a) QQ-plot for plink2. (b) QQ-plot for GMMAT with no kinship information (trivial diagonal only kinship matrix). (c) QQ-plot for GMMAT with kinship matrix inferred using SIGFRIED.
(PDF)

**S2 Fig. Illustration of the secure data aggregation protocol used in FedGMMAT.** Each site adds partitioned noise matrix to their respective partition. The encrypted matrices are pooled using round-robin schedule and sent the Central Server. Central Server decrypts the data matrix and sends respective partitions to each site. Sites locally remove the partitioned noise matrix from their partition and use the data.
(PDF)

**S3 Fig. Illustration of the OTP-like encryption for secure data aggregation protocol used in FedGMMAT.** The encryption in each site comprises adding a dense noise matrix that is generated using a secret seed that is shared with the central server. Note that each site adds partitioned noise matrix to their respective partition in addition to the server noise matrices. The pooled noise matrices are removed by the server from the aggregated data matrix.
(PDF)

**S4 Fig. The steps of null-model fitting stage.**
(PDF)

**S5 Fig. The steps of scoring stage.**
(PDF)

## Author Contributions

**Conceptualization:** Xiaoqian Jiang, Arif Harmanci.

**Data curation:** Wentao Li, Arif Harmanci.

**Formal analysis:** Wentao Li.

**Funding acquisition:** Han Chen, Xiaoqian Jiang, Arif Harmanci.

**Investigation:** Wentao Li.

**Methodology:** Wentao Li.

**Project administration:** Arif Harmanci.

**Resources:** Xiaoqian Jiang.

**Software:** Wentao Li.

**Supervision:** Arif Harmanci.

**Validation:** Wentao Li.

**Visualization:** Wentao Li, Arif Harmanci.

**Writing – original draft:** Wentao Li.

**Writing – review & editing:** Wentao Li, Han Chen, Xiaoqian Jiang, Arif Harmanci.

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
