## [Decision Letter · Decision Letter 0]

27 Dec 2023

Dear Dr. Harmanci,

Thank you very much for submitting your manuscript "FedGMMAT: Federated Generalized Linear Mixed Model Association Tests" for consideration at PLOS Computational Biology.

As with all papers reviewed by the journal, your manuscript was reviewed by members of the editorial board and by several independent reviewers. In light of the reviews (below this email), we would like to invite the resubmission of a significantly-revised version that takes into account the reviewers' comments.

We cannot make any decision about publication until we have seen the revised manuscript and your response to the reviewers' comments. Your revised manuscript is also likely to be sent to reviewers for further evaluation.

Sincerely,

William Stafford Noble

Section Editor

PLOS Computational Biology

William Noble

Section Editor

PLOS Computational Biology

Reviewer's Responses to Questions

**Comments to the Authors:**

Reviewer #1: In this manuscript, the author proposes a federated genetic association testing tool. As the genetic data can never be shared among collaborating sites due to the patient privacy protection regulation, the proposed method also is proposed to protect the intermediate statisticsby homomorphic encryption. Through three experiments utilizing both simulated and real-world datasets, the paper demonstrates the method's performance in comparison to the pooled analysis technique. The manuscript is notably well-structured and written in a clear manner. However, it is important to address several existing weaknesses within the content to enhance its overall quality.

Major comments:

1. For experiment 2, it would be helpful if the authors could explain the definition of homogeneous and heterogeneous. What kind of heterogeneity does the proposed method handle and how?

2. Can the proposed method also account for the random slope in the model?

3. The author built the federated testing procedure based on GLMM. It would be helpful if they discuss any potential limitations or constraints of this model, especially the rationale of using mixed-effect model. It would be also helpful if they explain the random effects in the data of the three experiements, and the potential causes of the existence of these random effects.

4. It would be good to see this algorithm to be utilized in a real-world collaborative study scenario. This would be good opportunity to demonstrate the validity and feasibility of the method’s utilization in a real multi-site collaboration.

5. This is a lossless distributed learning algorihtm for score test on genetic data.

Li R, Duan R, Zhang X, Lumley T, Pendergrass S, Bauer C, Hakonarson H, Carrell DS, Smoller JW, Wei WQ, Carroll R. Lossless integration of multiple electronic health records for identifying pleiotropy using summary statistics. Nature communications. 2021 Jan 8;12(1):168.

What’s the difference between your method and theirs?

6. How many iterations of updating model parameters (i.e., transferring aggregated information) are required in three experiements? It would be helpful if the author can discuss the feasibility of the implementation of this method in the real-world setting.

Minor comments:

1. On Page 3, I assume that HE means homomorphic encryption. However, this is not defined in the previous section or in the figure legend.

2. On Page 5, section B, the Appendix equation is not correctly referred.

3. On Page 7, section E Step 5, the equation is not correctly referred. Please check through the mansucript. There are other places with the same issue.

Reviewer #2: [Summary]

This paper presents FedGMMAT, a federated algorithm designed for generalized mixed model association tests. FedGMMAT extends the GMMAT algorithm by adapting it for federated analysis. This involves reconfiguring the algorithm to compute specific intermediate terms, including gradients, Hessian, and site-specific parameters, locally at each site. These terms are then securely aggregated across sites using homomorphic encryption. To facilitate this process, a trusted computing entity decrypts the aggregated statistics to perform global updates to the model. Additionally, FedGMMAT employs a two-step procedure for association testing. Initially, it trains a null model that includes only covariates. Subsequently, it calculates association p-values using score tests based on the null model. The effectiveness of the proposed algorithm is evaluated using three datasets: a small synthetic dataset (400 samples), the 1000 Genomes dataset with simulated phenotypes (6000 samples), and a real genotype-phenotype dataset from dbGaP (2545 samples). The results indicate that FedGMMAT produces comparable results to those obtained with GMMAT. Although this work offers technical insights into federated GLMMs, its practical applicability remains uncertain due to privacy protection and computational cost concerns.

[Major comments]

1. The practical viability of the proposed method in its intended settings is uncertain due to privacy issues associated with the exposure of intermediate statistics through FedGMMAT. Federating the analysis does not inherently ensure data protection. This manuscript makes exaggerated claims that may be misleading to readers. For instance, statements such as "[FL] complies with the regulations on PHI protection" and "intermediate statistics are protected by homomorphic encryption" are presented without adequate support. The manuscript lacks a thorough discussion of privacy protection provided by the proposed method or justification for the intermediate statistics that are disclosed, which is crucial for regulatory compliance and data protection:

- There seems to be an exposure of individuals' phenotype residuals (y_tilde) to the trusted computing entity through FedGMMAT. Given that many phenotypes are considered PHI, it raises the question of whether such a practice is feasible.

- Iterative updates to models, such as gradients and Hessian computations, may pose a risk of data leakage. This potential concern should be explored more comprehensively, and if possible, mitigations should be introduced and discussed.

- While the manuscript mentions the possibility of using other secure kinship estimation methods to compute the genetic kinship matrix (V) across sites, it does not address the feasibility of releasing kinship information for pairs of individuals between sites. It needs clarification whether FedGMMAT can still be applied when V cannot be disclosed.

2. The manuscript falls short in providing a comprehensive evaluation of the algorithm's runtime performance. Section II-D, which addresses communication cost performance, merely mentions that the size of the SNPs subset can impact computation time without discussing the applicability of the proposed method to large GWAS datasets or how the runtime scales with the number of samples. A particular concern arises from the proposed method's direct use of the genetic kinship matrix, which may result in limited scaling behavior.

3. The following prior work on federated mixed model association tests should be discussed and compared to the proposed method:

- Yan, Z., Zachrison, K. S., Schwamm, L. H., Estrada, J. J., & Duan, R. (2023). A privacy-preserving and computation-efficient federated algorithm for generalized linear mixed models to analyze correlated electronic health records data. PloS one, 18(1), e0280192.

- Li, W., Chen, H., Jiang, X., & Harmanci, A. (2023). Federated generalized linear mixed models for collaborative genome-wide association studies. Iscience, 26(8).

- Chen, J., Edupalli, M., Berger, B., & Cho, H. (2022). Secure and federated linear mixed model association tests. bioRxiv, 2022-05.

- Zhu, R., Jiang, C., Wang, X., Wang, S., Zheng, H., & Tang, H. (2020). Privacy-preserving construction of generalized linear mixed model for biomedical computation. Bioinformatics, 36(Supplement_1), i128-i135.

[Minor comments]

1. The manuscript has many grammatical errors. The authors need to thoroughly revise the writing.

2. While it is emphasized that "null model fitting does not utilize any of the sensitive genotype data," it is important to note that the covariate and phenotype data are also sensitive.

3. Round-robin updates appear less efficient compared with the approach of calculating all updates simultaneously and aggregating them to a single site.

4. Section IV-C (Parameters estimation) is difficult to follow as it lists the terms to be computed without providing explanations for each.

5. To assess FedGMMAT's correction for kinship, it would be beneficial to include results from non-GMM association tests for comparison.

**Have the authors made all data and (if applicable) computational code underlying the findings in their manuscript fully available?**

Reviewer #1: Yes

Reviewer #2: Yes

PLOS authors have the option to publish the peer review history of their article (what does this mean?). If published, this will include your full peer review and any attached files.

Reviewer #1: No

Reviewer #2: No
---

## [Decision Letter · Decision Letter 1]

7 May 2024

Dear Dr. Harmanci,

We are pleased to inform you that your manuscript 'FedGMMAT: Federated Generalized Linear Mixed Model Association Tests' has been provisionally accepted for publication in PLOS Computational Biology.

Best regards,

Shihua Zhang

Section Editor

PLOS Computational Biology

William Noble

Section Editor

PLOS Computational Biology

Reviewer's Responses to Questions

**Comments to the Authors:**

Reviewer #1: Thank you for fully addressing my comments. I believe the manuscript has improved and wish you the best for its forthcoming publication process in the journal.

**Have the authors made all data and (if applicable) computational code underlying the findings in their manuscript fully available?**

Reviewer #1: Yes

PLOS authors have the option to publish the peer review history of their article (what does this mean?). If published, this will include your full peer review and any attached files.

Reviewer #1: No

---

## [Editor Report · Acceptance letter]

18 Jul 2024

PCOMPBIOL-D-23-01279R1 

FedGMMAT: Federated Generalized Linear Mixed Model Association Tests

Dear Dr Harmanci,

I am pleased to inform you that your manuscript has been formally accepted for publication in PLOS Computational Biology. Your manuscript is now with our production department and you will be notified of the publication date in due course.

With kind regards,

Anita Estes
